# Graphein - a Python Library for Geometric Deep Learning and Network Analysis on Biomolecular Structures and Interaction Networks

**Arian R. Jamasb**[1,2]*, **Ramon Viñas**[2], **Eric J. Ma**[3], **Charlie Harris**[1,2], **Kexin Huang**[4],
**Dominic Hall**[1], **Pietro Lió**[2]*, **Tom L. Blundell**[1]
[1] Department of Biochemistry, University of Cambridge
[2] Department of Computer Science & Technology, University of Cambridge
[3] PyMC Labs
[4] Department of Computer Science, Stanford University

## Abstract

Geometric deep learning has broad applications in biology, a domain where relational structure in data is often intrinsic to modelling the underlying phenomena. Currently, efforts in both geometric deep learning and, more broadly, deep learning applied to biomolecular tasks have been hampered by a scarcity of appropriate datasets accessible to domain specialists and machine learning researchers alike. To address this, we introduce Graphein as a turn-key tool for transforming raw data from widely-used bioinformatics databases into machine learning-ready datasets in a high-throughput and flexible manner. Graphein is a Python library for constructing graph and surface-mesh representations of biomolecular structures, such as proteins, nucleic acids and small molecules, and biological interaction networks for computational analysis and machine learning. Graphein provides utilities for data retrieval from widely-used bioinformatics databases for structural data, including the Protein Data Bank, the AlphaFold Structure Database, chemical data from ZINC and ChEMBL, and for biomolecular interaction networks from STRINGdb, BioGrid, TRRUST and RegNetwork. The library interfaces with popular geometric deep learning libraries: DGL, Jraph, PyTorch Geometric and PyTorch3D though remains framework agnostic as it is built on top of the PyData ecosystem to enable inter-operability with scientific computing tools and libraries. Graphein is designed to be highly flexible, allowing the user to specify each step of the data preparation, scalable to facilitate working with large protein complexes and interaction graphs, and contains useful pre-processing tools for preparing experimental files. Graphein facilitates network-based, graph-theoretic and topological analyses of structural and interaction datasets in a high-throughput manner. We envision that Graphein will facilitate developments in computational biology, graph representation learning and drug discovery.

**Availability and implementation**: Graphein is written in Python. Source code, example usage and tutorials, datasets, and documentation are made freely available under the MIT License at the following URL: https://graphein.ai

---

*To whom correspondence should be addressed: `arj39@cam.ac.uk`, `pl219@cl.cam.ac.uk`

36th Conference on Neural Information Processing Systems (NeurIPS 2022).

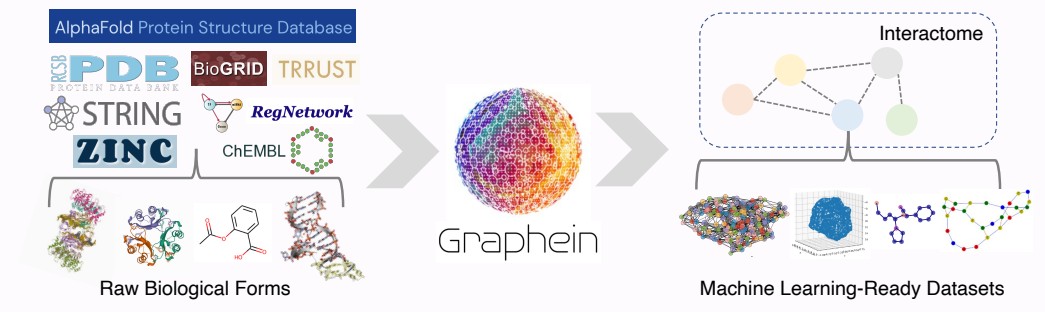

Figure 1: Graphein rapidly transforms and integrates raw biological and chemical data into actionable machine learning-ready datasets.

# 1 Introduction

The functional roles of macromolecules are intricately tied to the complex three dimensional structures they adopt. Many biological functions are mediated by interacting biomolecular entities, often through direct physical contacts governed by their 3D structures. Recent developments in protein folding have given rise to an explosion of structural data available to machine learning practitioners [1, 2]. Highly-accurate protein structure prediction using AlphaFold2 has been applied at the proteome scale to humans and 20 key model organisms [1, 3]. Recent work making use of these data have demonstrated the power of GNNs applied to protein structures methods over models trained on large corpora of sequences alone [4] for protein representation learning and highlighted the importance of carefully considering the graph representation. We anticipate a significant growth in the availability of biomolecular structural and interaction data in the coming years as both computational and experimental techniques mature. In particular, we identify structural interactomics as an emerging application area for geometric deep learning as sparse experimental structural coverage of interactomes can be annotated with modelled macromolecular structures. However, the question of how to best leverage and integrate these data with other modalities remains. A recent review of biomedical knowledge graph datasets identifies graph composition, feature and metadata incorporation and reproducibility as key challenges [5]; we believe these are vital considerations for structural and interactomic data. To address these issues, we present Graphein, a flexible tool that provides greater control over the data engineering and featurisation process of structural data, abstracting away cumbersome data preparation tasks and facilitating reproducible research.

Graphein provides a bridge for geometric deep learning into structural interactomics through convenient structure retrieval and graph creation tools. Our library interfaces with several protein structure databases to leverage decades of structural biology research as well as recent developments in protein folding. These advances have resulted in a large pool of experimentally-determined and modelled protein structures, with massive potential to inform future research [6, 1]. To realise this potential, Graphein additionally represents these graphs at different levels of granularity, including atom, residue, secondary structure and chain-level graphs, and populates the corresponding node and edge attributes with informative features, enabling a wide range of downstream applications of geometric deep learning.

**Representing Structural Biomolecular Data**    It is not yet clear how best to represent these data in machine learning experiments. It has been shown that structure-based methods frequently outperform sequence-based methods and that the choice of architecture shows strong task dependency [7]. 3D Convolutional Neural Networks (3DCNNs) have been routinely applied to grid-structured representations of protein structures and sequence-based methods have proved commonplace [8, 9, 10]. Surface-based methods have proved effective for predicting protein-protein and protein-ligand interactions [11, 12]. Nonetheless, these representations fail to capture relational information in the context of intramolecular contacts and the internal chemistry of the biomolecular structures. Furthermore, these methods are often computationally inefficient due to convolving over large regions of empty space, and computational constraints often require the volume of the protein considered to regions of interest, thereby losing global structural information. For instance, in the case of protein-ligand interaction and binding affinity prediction, central tasks in data-driven drug discovery, this often takes

the form of restricting the volume to be centred on a binding pocket, thereby losing information about allosteric sites on the protein and possible conformational rearrangements that contribute to molecular recognition. Furthermore, 3D volumetric representations are not rotationally invariant, a deficiency that is often mitigated using costly data augmentation techniques. Graphs suffer relatively less from these problems as they are translationally and rotationally invariant. Structural descriptors of position can be leveraged and meaningfully exploited by architectures such as Equivariant Neural Networks (ENNs), which ensure geometric transformations applied to their inputs correspond to well-defined transformations of the outputs.

Macromolecular structures and biological interaction networks can naturally be represented as graphs at different levels of granularity and abstraction with edges denoting intramolecular, regulatory and functional interactions or spatial relationships. The graph structure can further be elaborated by assigning metadata and numerical features to nodes and edges as well as the whole graph. These features can represent, for instance, chemical properties of residues or atoms, secondary structure assignments or solvent accessibility metrics of the residue or descriptions of geometry. Edge features can include bond or interaction types as well as distances. Graph features can include functional annotations or sequence-based descriptors. In the context of interaction networks, structural data can be overlaid on protein nodes providing a multi-scale view of biological systems and function. By providing a toolkit for integrating data across these domains Graphein provides a bridge for geometric deep learning into structural interactomics.

## 2 Related Work

**Geometric Deep Learning Tools**   Geometric deep learning methods have demonstrated their suitability for tasks across domains. In part, this has been fuelled by the development of libraries that provide easy access to non-Euclidean data objects and models from the literature. Deep Graph Library (DGL) [13] and PyTorch Geometric [14] are the main open-source frameworks built for PyTorch [15]. Other tools include: Graph Nets [16] for Sonnet [17]/Tensorflow [18] and Jraph [19] for JAX [20]. Whilst several of these provide datasets and simple featurisation schemes relevant to the life sciences, these are typically focussed on small molecules [13, 14]. However, data preparation for geometric deep learning in structural biology and interactomics is yet to receive the same attention and tools addressing this bottleneck can greatly accelerate research in these domains [21].

**Datasets and Benchmarks**   In-built dataset support is a common feature of geometric deep learning frameworks. More specialised libraries, such as DGL-LifeSci, DeepChem and TorchDrug, provide datasets, featurisation, neural network layers and pre-trained models for tasks involving small molecules in the life sciences, computational chemistry and drug development [22, 23, 24]. TorchDrug and DeepChem provide reinforcement learning environments to fine tune generative models for physicochemical properties such as drug-likeness (QED) and lipophilicity (LogP). Therapeutics Data Commons provides raw datasets for small molecule and biologics tasks [25]. However, none of these tools address the transformation of macromolecular structures into machine learning-ready formats or provide comprehensive utilities for the generation of new datasets.

Biomolecular tasks are included in many graph representation learning benchmarks. The Open Graph Benchmark (OGB) includes graph property prediction tasks on small molecules, link prediction tasks (ogbl-ppa) based on protein-protein interaction prediction and a biomedical knowledge graph (ogbl-biokg), and a node classification task based on prediction of protein function (ogbn-proteins) [26]. The TUDataset contains three biologically-motivated benchmark datasets for graph classification, (PROTEINS, ENZYMES and DD) relevant to applications in structural biology [27]. For PROTEINS and DD the goal is to predict whether or not a protein is an enzyme and these are derived from the same data under differing graph construction schemes [28, 29]. ENZYMES provides a task based on assigning Enzyme Commission (EC) numbers to graph representations of enzyme structures derived from the BRENDA database [30]. However, these collections have been abstracted away from the underlying biological data and computing additional features and metadata is cumbersome. More recently, ATOM3D provides a collection of benchmark datasets for structurally-motivated tasks on biomolecules and show leveraging structural information consistently improves performance, and that the choice of architecture significantly impacts performance depending on the task context [7].

**Tools for Working with Biomolecular Graphs** Whilst tools exist for transforming protein structures into graphs, they typically focus on visualisation and traditional bioinformatics analyses for specific use-cases and cannot be conveniently used for developing deep learning models [31]. GraProStr is a web-server that enables users to submit structures for conversion into a graph which can be downloaded as textfiles [32]. This provides users with limited control over the construction process, low-throughput and limited featurisation support. Furthermore GraProStr provides no utilities for machine learning or unifying structural and interactomic data. Mayavi, and GSP4PDB & LIGPLOT provide utilities for visualising protein structures and protein-ligand interaction as graphs, respectively [33, 34, 35]. Bionoi is a library for representing protein-ligand interactions as voronoi diagram images specifically for applications in machine learning [36]. PyUUL is a recent tool designed for transforming biological structures into formats suitable for computer vision tasks, such as voxels and point clouds. However, graph-based representations and interaction data are not addressed [21]. The lack of fine-grained control over the construction and featurisation, public APIs for high-throughput programmatic access, ease of integrating data modalities, and incompatibility with deep learning libraries motivated the development of Graphein as machine learning-first library.

## 3 Graphein

Graphein provides utilities for constructing geometric representations of protein and RNA structures, small moleculure, protein-protein interaction networks, and gene regulatory networks. The library provides high- and low-level APIs, appropriate for both novice and experienced users. The high-level API constructs geometric representations of structural and interaction data from standard biological identifiers. The low-level API offers a fine-grained customisation of the graph selection from the input data, allowing users to define their own data preparation, graph construction and featurisation functions in a consistent manner. Graphein is built on the PyData Stack to allow for easy inter-operability with standard scientific computing tools and deep learning framework agnosticism. Graphein is organised into submodules for each of the supported modalities (Figure 2).

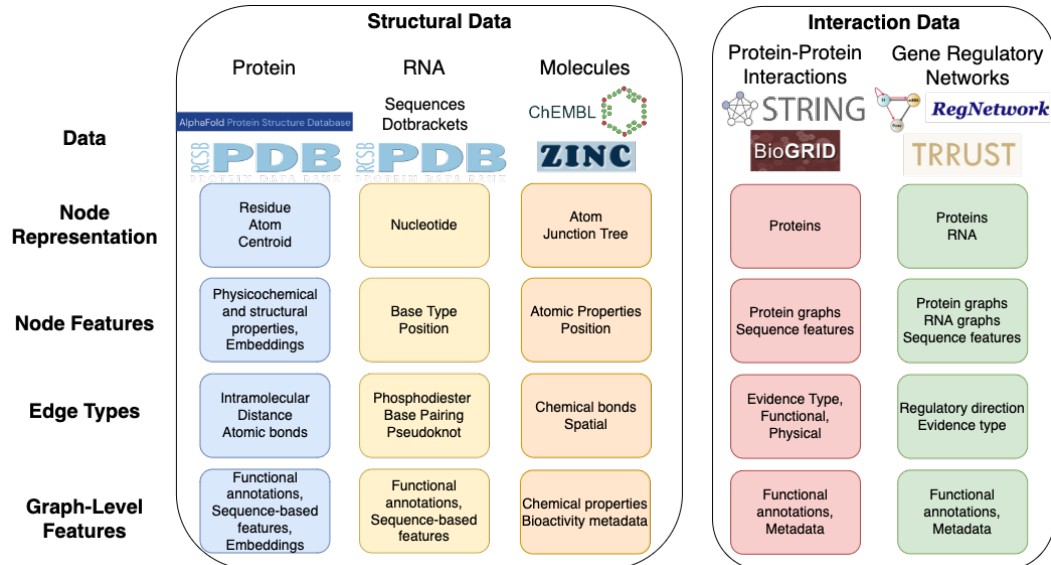

Figure 2: Overview of graph and mesh construction and featurisation schemes for data modalities supported by Graphein. Modules are inter-operable allowing protein or RNA structure graph construction to be applied to nodes in regulatory networks.

### 3.1 Protein Structure Graphs

Graphein interfaces with the PDB and the AlphaFold Structure Database to create geometric representations of protein structures. Furthermore, users can supply their own .pdb files, enabling pre-processing with standard bioinformatics tools and pipelines. An overview of featurisation schemes is provided in Supplementary Information A.

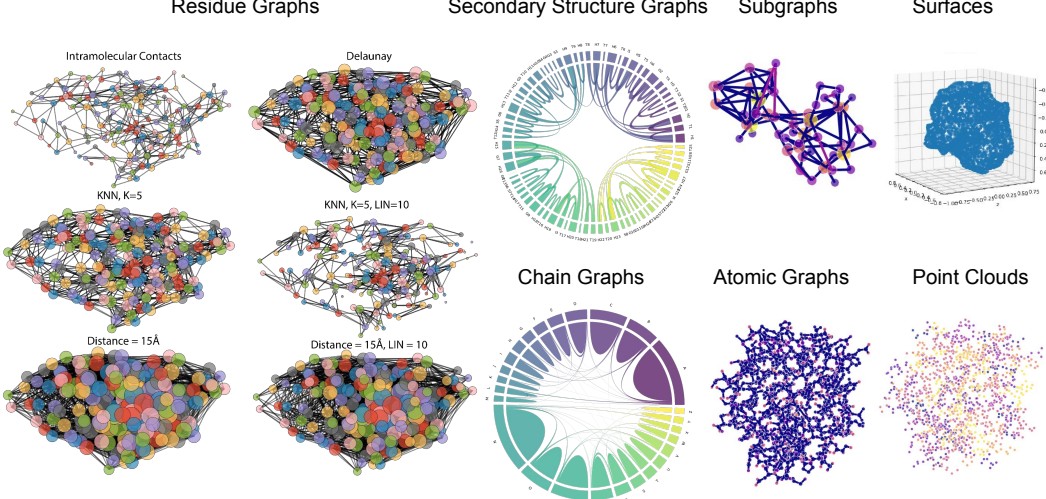

Figure 3: Representation types for structural data processed with Graphein. Plots generated with Graphein's visualisation tools.

**Node Representations**   Graphs can be constructed for all chains contained within a polypeptide structure, or for a user-defined selection of chains. This is useful in contexts where regions of interest on a protein may be localised to a single chain. For residue-level graphs, users can choose between atom-based positional information, or sidechain centroids. Sidechain centroids are calculated as the centre of gravity of the deprotonated residue. Residue-level graph nodes can be featurised with a one-hot encoding of amino acid type, physicochemical and biochemical properties retrieved from the ExPaSY ProtScale [37] which includes 61 descriptors such as iso-electric points, mutability and transmembrane tendencies. Additional numerical features can be retrieved from AAIndex [38]. Low-dimensional embeddings of amino acid physicochemical properties are provided from Kidera et al. [39] and Meiler et al [40]. In addition to fixed embeddings, sequence embeddings can be retrieved from large pre-trained language models, such as the ESM-1b Transformer model [41] and BioVec [42]. Secondary structural information can be included via a one-hot encoded representation of eight state secondary structure and solvent accessibility metrics (ASA, RSA, SSA) computed by DSSP [43]. $x, y, z$ positions are added as node features. Vector-based features capturing important aspects of protein structure, such as sidechain position and dynamics derived from Normal Mode Analysis, are provided. These have been shown to be highly informative features [44] and have the potential to be meaningfully incorporated in equivariant neural network models. Functionality for user-defined node or edge features is also provided with useful utilities allowing for computation or aggregation of features over constituent chains. Figure 2 illustrates an overview of the mesh and graph construction methods as well as the node and edge featurisation schemes; Figure 3 shows example visualisations of graph and meshes produced by Graphein.

**Edge Representations**   Graphein provides utilities in the high-level API for a number of edge-construction schemes. The low-level API provides a simple and intuitive way for users to define novel edge construction schemes. Edge construction methods are organised into distance-based, intramolecular interaction-based, and atomic structure-based submodules. Each of these edge construction methods are composable to produce multirelational graphs. This is particularly useful for models that operate on different levels to capture varying aspects of the underlying network.

Functionality for computing intramolecular graph edges is provided through distance-based heuristics as well as through an optional dependency, GetContacts [45]. Euclidean distance-based edges can be computed with a user-defined threshold. Functionality for constructing $k$-nearest neighbour graphs, where two vertices are connected by an edge if they are among the $k$-nearest neighbours by Euclidean distance is included. Graph edges can also be added on the basis of the Delaunay triangulation. Delaunay triangles correspond to joining points that share a face in the 3D Voronoi diagram of the protein structures. For distance-based edges, a Long Interaction Network (LIN) parameter controls the minimum required separation in the amino acid sequence for edge creation. This is useful to

reduce the number of noisy edges under distance-based edge creation schemes. Edge featurisation for atom-level graphs is provided by annotations of bond type and ring status.

**Protein-Ligand Graphs & Subgraphs**     Graphein provides powerful selection methods for extracting subgraphs from regions of interest within macromolecular structures. Proteins are often large and the region of the structure relevant to a given task may be localised to a region of the structure. For instance, we can extract binding pocket subgraphs based on a known ligand pose as well as subgraphs of interfaces in the case of complexed structures, or surface-subgraphs that retain only solvent-exposed residues.

## 3.2   Protein Structure Meshes

Geometric deep learning applied to surface representations of protein structures have demonstrated promise on a variety of tasks in the context of structural biology and structural interactomics [11, 46]. The protein structure mesh module consists of a wrapper for PyMOL, a commonplace molecular informatics visualisation tool, and Pytorch3D [47]. PyMol is used to produce a `.Obj` file from either a PDB accession code or a provided `.PDB` file, enabling the use of pre-processed structures. Pytorch3D is used to produce a tensor-based representation of the protein surface as vertices and faces. Users can pass any desired parameters or commands controlling the surface calculation to PyMol via a configuration object. These parameters include specifying solvent inclusion, solvent probe radius, surface type, and resolution. We provide sane defaults for first-time users. To our knowledge, this is the first application of PyTorch3D for protein structure data.

## 3.3   Molecules

Featurisation schemes and computational analysis of small molecular graphs are a relatively mature area of research. Typically, nodes represent atoms and edges denote the chemical bonding structure between them. Junction trees are a compact representation of molecular graphs, coarsening the graph such that nodes represent molecular substructures [48]. Graphein provides utilities for working with both of these representations and conformer generation to enable models operating on molecules embedding in 3D space which has been shown to be highly effective for representation learning [49]. Graphein provides extensive featurisation options for molecules and also supports working with molecular fragments enabling workflows in fragment-based drug design. Molecular structure data can be retrieved from both ZINC [50] and ChEMBL [51]. Furthermore, the wealth of curated metadata, such as bioactivity, from ChEMBL can be queried and used in featurisation.

## 3.4   RNA Structures

Ribonucleic Acid (RNA) is a nucleotide biopolymer capable of forming higher-order structural arrangements through self-association mediated by complementary base pairing interactions. Graphein provides utilities for constructing secondary structure graphs and 3D of RNA structures, taking as input a crystallographic structure or an RNA sequence and an associated string representation of the secondary structure in dotbracket notation [52]. Graphs can be constructed using two types of bonding between nucleotides: phosphodiester bonds between adjacent bases, and base-pairing interactions between complementary bases specified by the dotbracket string (Figure 4). Graphein also supports addition of pseudoknots - structural motifs composed of interactions between intercalated hairpin loops specified in the dotbracket notation.

## 3.5   Interaction Networks

Interactomics presents a clear application of geometric deep learning as these data are fundamentally relational in structure. Biomolecular entities can be represented as nodes, and their associated functional relationships and physical interactions can be represented as edges with associated metadata, such as the direction and nature of regulation. For a full discussion of applications, datasets and modelling techniques we refer readers to the reviews by [5] and [53]. Graphein implements interaction graph construction from protein-protein interaction and gene regulatory network databases. Interaction graphs integrate networks from several sources and can be constructed in a highly customisable way (see Supplementary Information B, C for a summary of user-definable parameters) with the ability to overlay structural models for the biomolecular species involved.

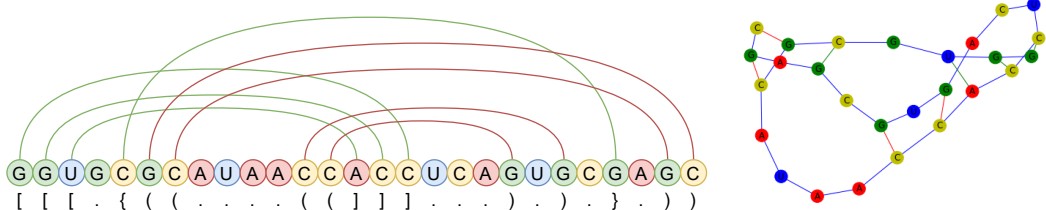

Figure 4: Example RNA Secondary Structure Graph. RNA Secondary structures can be represented as dotbracket strings and multi-relational graphs. Blue edges indicate phosphodiester backbone linkages, red edges indicate base-pairing interactions and green edges indicate pseudoknot pairings.

### 3.5.1 Protein-Protein Interaction Networks

Many of the functional roles of proteins are carried out by larger assemblies of protein complexes and many biological processes are regulated through interactions mediated by physical contacts. Understanding these functions is central to characterising healthy and diseased states of biological systems. Graphein interfaces with widely-used databases of biomolecular interaction data for easy retrieval and construction of graph-based representations of protein-protein interactions.

**STRING** is a database of more than 20 billion known and predicted functional and direct physical protein-protein interactions between 67.6 million proteins across 14,094 organisms [54]. Predicted interactions in STRING are derived from genomic context, high-throughput experimental procedures, conservation of co-expression, text-mining of the literature and aggregation from other databases. STRING is made freely available by the original authors under a Creative Commons BY 4.0 license.

**BioGRID** is a database that archives protein interaction data from model organisms and humans, curated from high-throughput studies and individual studies. The database contains 2,127,726 protein and genetic interactions curated from 77,696 publications [55]. BioGRID is made available for academic and commercial use by the original authors under the MIT License.

### 3.5.2 Gene Regulatory Networks

Gene regulatory networks (GRNs) consist of collections of genes, transcription factors (TFs) and other regulatory elements, and their associated regulatory interactions. Reconstructing transcriptional regulatory networks is a long-standing problem in computational biology in its own right due to its relevance to characterising healthy and diseased states of cells, and these data can provide meaningful signal in other contexts such as multi-modal modelling of biological systems and phenomena. Graphein supports GRN graph construction from two widely-used databases, allowing users to easily unify datasets and construct graph representations of these networks.

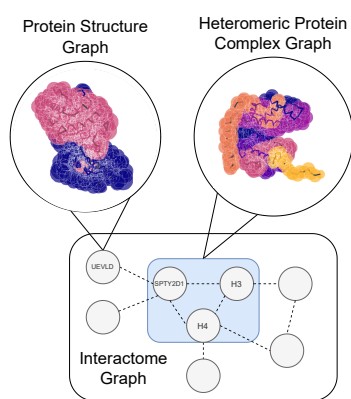

Figure 5: Graphein can facilitate the integration of structural and biomolecular interaction data to enable geometric deep learning research in structural interactomics. 3D visualisations of graphs are generated using Graphein.

**TRRUST** is a database of regulatory interactions for human and mouse interactomes curated from the literature via a sentence-based text-mining approach [56]. The current release contains 8,427 / 6,490 regulatory interactions with associated regulatory directions (activation/repression) over 795 / 827 transcription factors and 2,067 / 1,629 non-transcription factor genes for humans and mice, respectively. TRRUST is made freely available by the original authors for non-commercial research under a Creative Commons Attribution-ShareAlike 4.0 International License.

**RegNetwork** is a database of transcription-factor and miRNA mediated regulatory interactions for humans and mice [57]. RegNetwork is an aggregation of 25 source databases from which the regulatory network is populated and annotated. The latest release contains 14,981 / 94,876 TF-gene,

361 / 129 TF-TF, 21,744 / 25,574 TF-miRNA, 171,477 / 176,512 miRNA-gene and 25,854 / 26,545 miRNA-TF interactions over 1,456 / 1,328 transcription factors, 1,904 / 1,290 miRNAs and 19,719 / 18,120 genes for humans and mice, respectively. The dataset is made publicly available by the original authors.

## 4 Datasets

As example workflows, we make available two graph-based protein structure datasets focussed on tasks where relational inductive biases appear intuitively useful and demonstrate how Graphein can help formulate different tasks from the same underlying dataset.

**PPISP - Protein Protein Interaction Site Prediction**    The first, based on the collections outlined in [58], consists of 420 protein structures, with node labels indicating whether a residue is involved in a protein-protein interaction - a task central to structural interactomics [59]. The data originate from co-crystallised structures of the complexes in the RCSB PDB. The authors make available a set of additional node features based on Position-Specific Scoring Matrices (PSSMs), providing evolutionary context as to which protein-protein interaction sites are typically conserved, which can be incorporated with the structural node features calculated by Graphein. This dataset was used in [60] in conjunction with Graphein to compute the protein structure graph inputs to a Message-Passing Neural Process model which achieved state-of-the-art performance.

**PSCDB - Protein Structural Change Database**    The second dataset, based on Protein Structural Change Database (PSCDB) [61], consists of 904 paired examples of bound and unbound protein structures that undergo 7 classes of conformational rearrangement motion. Prediction of conformational rearrangement upon ligand binding is a longstanding problem in computational structural biology and has significant implications for drug discovery and development. Two tasks can be formulated with this dataset. The first is the graph classification task of predicting the type of motion a protein undergoes upon ligand binding, the second is framing prediction of the rearrangement itself as an edge prediction task between the paired bound and unbound protein structure graphs. These tasks provide utility in improving understanding of protein structural dynamics in drug development, where molecules are typically docked into largely rigid structures with limited flexibility in the binding pockets in high-throughput *in silico* screens. PSCDB is made publicly available by the original authors and we provide a processed version in our repository.

**ccPDB**    We derive four datasets, each with a graph and a node classification task from the ccPDB [62]. The ccPDB provides collections of protein structures and annotations of interactions with various molecular species. The proteins are high-quality, non-redundant sets (25% sequence identity) with maximum resolution of 3 Å, minimum sequence length of 80 residues. Node-level annotations of interaction are provided in each case with the cutoff set at 4 Å. ccPDB is made freely available online.

- `PROTEINS_METAL` contains protein structures that bind 7 types of metal ions (Fe, Mg, Ca, Mn, Zn, Co, Ni; $n = 215 / 1,908 / 1,402 / 521 / 1,660 / 201 / 355$).
- `PROTEINS_NUCLEOTIDES` contains protein structures that bind 8 species of nucleotides (ATP, ADP, GTP, GDP, NAD, FAD, FMN, UDP; $n = 313 / 353 / 83 / 120 / 140 / 172 / 117 / 68$)
- `PROTEINS_NUCLEIC` contains protein structures that bind DNA or RNA polymers ($n = 560 / 415$).
- `PROTEINS_LIGAND` contains protein structures that bind 7 species of ligands ($SO_4$, $PO_4$, NAG, HEM, BME, EDO, PLP; $n = 3312 / 1299 / 727 / 176 / 191 / 1507 / 65$).

## 5 Machine Learning Utilities

Here we introduce the different machine learning utilities that Graphein provides. They include modules to convert graph object into machine learning-ready formats, manipulate and visualise graphs, appropriately split datasets to avoid data-leaking, several usage tutorials and template notebooks.

**Conversion & Dataset Classes**   Convenience utilities for converting between NetworkX [63] graph objects and commonly-used geometric deep learning library data objects are provided for DGL, PyTorch Geometric and Jraph. Underlying graph objects are based on NetworkX, enabling conversion to other formats. We provide wrapped PyTorch Geometric dataset classes which handle downloading of structures and preparation of graphs. As the required user inputs are a list of identifiers, labels (optionally) and a Graphein config, these are very lightweight formats for sharing entire pre-processing pipelines that can be readily reused and adapted by others.

**Adjacency Tensors, Diffusion Matrices & Line Graphs**   Graphein provides utilities for computation of diffusion matrices (and related adjacency matrices) to (1) facilitate exploration of biological data with models that leverage these representations, and (2) aid in the construction of diffusion matrices for graph neural networks. We also provide utilities for computing line graphs. Performing edge message passing on line graphs [64] has been shown to be highly effective in representation learning on 3D molecules and protein structures [4].

**Visualisation**   Built-in interactive tools are provided for each of the modalities supported to allow inspection of data in pre and post-processing. Further utilities for analysing and plotting graph properties are provided.

**Data Splitting**   Splitting datasets of protein structures requires some consideration to avoid leaking data through homologous proteins as proteins with low-levels of sequence identity can adopt very similar folds. In most cases (i.e. where the data have not already been filtered for redundancy), random sampling is unsuitable as the data are unlikely to be independent due to evolutionary relationships. We provide utilities for splitting and clustering datasets based on sequence homology using BLAST.

## 6   Usage

Example usage and workflows are provided in the documentation. Examples and tutorials are provided as runnable notebooks detailing use of the high and low-level APIs for the data modalities currently supported by Graphein, and the ease of ingesting novel structural datasets into a suite of geometric deep learning models. Source code is made at: `https://graphein.ai`.

## 7   Conclusion

Geometric deep learning has shown promise in computational biology and structural biology. However, the availability of processed datasets is a research bottleneck. Graphein is a Python library designed to facilitate construction of datasets for geometric deep learning applied to biomolecular structures and interactions. By providing tools for these modalities, we hope to facilitate research in data-driven structural interactomics. We make available two example datasets for protein-protein interaction site prediction (node classification) and protein conformational rearrangement prediction (graph classification and edge prediction) as well as extensive documentation and tutorial notebooks.

Whilst graphs are a natural representation of biological interaction data, hypergraphs may provide a higher-fidelity representation of the underlying biological relationships. Many interactions are contextual, or require multiple obligate interactors, which can be represented by hyperedges between several entities required for a functional or structural relationship. We are also pursuing richer representations of dynamics, both in structural data and in interactions as these are central biological components that are beyond the scope of the present release. These features will be included in subsequent releases and the API design of Graphein makes it simple for users to write and contribute their own workflows. Graphein implements a high-level and low-level API to enable rapid and fine-grained control of data preparation. Graphein is provided as Free Open Source Software under a permissive MIT License which we hope will encourage the community to contribute customised workflows to the library. We hope that Graphein serves to further progress in the field and reduce friction in processing structural and interaction data for geometric deep learning. The library also provides utility in preparing protein structure and interactomics graphs for graph-theoretic and topological data analyses which we hope will draw further insight from the computational biology community.

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
