# Graphein - Supplementary material

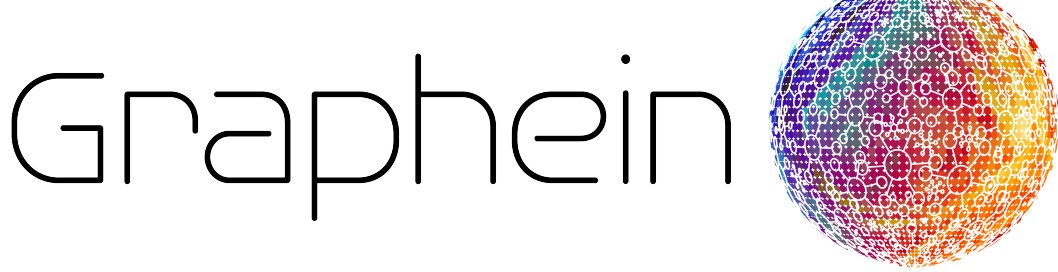

35th Conference on Neural Information Processing Systems (NeurIPS 2021) Track on Datasets and Benchmarks.

# Contents

# A   Featurisation Schemes for Protein Structure Graphs

Table 1: Geometric representation options for a protein structure

| Node Features | | |
| --- | --- | --- |
| Node Type | Feature | Source |
| **Residue** | | |
| | Molecular Weight | |
| | $x, y, z$ Co-ordinates | |
| | Sidechain Vector | |
| | Neighbour Vectors | |
| | $\beta$ Carbon Vectors | |
| | $\phi$ Torsion Angle | DSSP [1] |
| | $\psi$ Torsion Angle | DSSP [1] |
| | Secondary Structure | DSSP [1] |
| | Solvent Accessibility | DDSP [1] |
| | Low-dimensional embeddings of physicochemical properties | Meiler et al. [2] |
| | ExPaSy Protein Scale | [3] |
| | AAIndex Descriptors (various) | [4] |
| | ESM Transformer Protein Language Model Embedding | ESM [5] |
| | BioVec Protein Language Model Embedding | ProtVec [6] |
| **Atom** | | |
| | Atomic Weight | |
| | Covalent Radius | [7] |
| | H-bond Donor Status | |
| | H-bond Acceptor Status | |
| Edge Types | | |
| Node Type | Edge Type | Source |
| **Atom** | | |
| | Covalent Bonds | |
| **Residue** | | |
| | Hydrophobic Interactions | |
| | Disulfide Interactions | |
| | Hydrogen Bonds | |
| | Ionic Interactions | |
| | Aromatic Interactions | |
| | Aromatic-Sulphur Interactions | |
| | Cation-$\pi$ Interactions | |
| | Peptide Bonds | |
| | $\pi$ Stacking Interactions | [8] |
| | Salt Bridge | [8] |
| | t Stacking | [8] |
| | Van der Waals | [8] |
| **Any** | | |
| | K-Nearest Neighbours | |
| | Delaunay Triangulation | |
| | Distance Threshold | |
| | Distance Window | |
| | Sequence Distance | |
| Graph-level Features | | |
| | Features | Source |
| **Sequence** | | |
| | Molecular Weight | |
| | Transformer Positional Encoding | |
| | ESM Transformer Protein Language Model Embedding | ESM [5] |
| | BioVec Protein Language Model Embedding | ProtVec [6] |
| | Amino Acid Composition | ProPy [9] |
| | Dipeptide Composition | ProPy [9] |
| | Tripeptide Composition | ProPy [9] |
| | Moreau-Broto Autocorrelation | ProPy [9] |
| | Moran Autocorrelation | ProPy [9] |
| | Geary Autocorrelation | ProPy [9] |
| | Sequence-order-coupling Number | ProPy [9] |
| | Quasi-Sequence-Order Descriptors | ProPy [9] |
| | CTD Descriptors | ProPy [9] |

# B   Featurisation Schemes for RNA Structure Graphs

Table 2: Geometric representation options for a RNA structure

| Node Type | Feature | Source |
|---|---|---|
| *Node Features* | | |
| Node Type | Feature | Source |
| **Residue** | | |
| | Molecular Weight | |
| **Atom** | $x, y, z$ Co-ordinates | |
| | Atomic Weight | |
| | Covalent Radius | [7] |
| *Edge Types* | | |
| Node Type | Edge Type | Source |
| **Atom** | | |
| | Covalent Bonds | |
| | K-Nearest Neighbours | |
| | Delaunay Triangulation | |
| | Distance Threshold | |
| | Distance Window | |
| | Sequence Distance | |
| **Base** | | |
| | Phosphodiester Bonds | |
| | Base Pairing Interactions | |
| | Pseduoknots | |
| *Graph-level Features* | | |
| | Features | Source |
| **Sequence** | | |
| | Molecular Weight | |
| | Transformer Positional Encoding | |

# C  Featurisation Schemes for Molecular Graphs

Table 3: Geometric representation options for a molecular graph

| Node Features | | |
|---|---|---|
| Node Type | Feature | Source |
| **Atom/Junction Tree** | | |
| | Atomic Mass | RDKit |
| | Covalent Radius | |
| | Atom Type | |
| | Covalent Degree | |
| | Total Valence | RDKit |
| | Explicit Valence | RDKit |
| | Implicit Valence | RDKit |
| | Implicit Hydrogens | RDKit |
| | Explicit Hydrogens | RDKit |
| | Total Hydrogens | RDKit |
| | Radical Electrons | RDKit |
| | Formal Charge | RDKit |
| | Hybridization | RDKit |
| | Aromatic Status | RDKit |
| | Ring Status | RDKit |
| | Ring Size X Status | |
| | Isotope Status | RDKit |
| | Formal Charge | RDKit |
| | Chiral Status | RDKit |
| *Edge Types* | | |
| Node Type | Edge Type | Source |
| **Atom/Junction Tree** | | |
| | Covalent Bonds | |
| | K-Nearest Neighbours | |
| | Delaunay Triangulation | |
| | Distance Threshold | |
| | Distance Window | |
| | Fully Connected | |
| *Edge Features* | | |
| | Feature | Source |
| | Bond Order | RDKit |
| | Aromatic Status | RDKit |
| | Conjugation Status | RDKit |
| | Ring Status | RDKit |
| | Ring Size X Status | |
| | Stereo Configuration | RDKit |
| *Graph-level Features* | | |
| | Features | Source |
| | ChEMBL Metadata | [10] |
| | Molecular Weight RDKit | |
| | Geometric Center | RDKit |
| | Principal Moments of Inertia | |
| | Max Ring Size | |
| | Morgan Fingerprint | RDKit |
| | Fragment Counts | |
| | QED Score | RDKit |

# D Parameters for protein-protein interaction graphs

## D.1 General parameters

The following table shows the generic Graphein parameters for protein-protein interaction graphs.

| Parameter | Default | Valid values | Description |
|---|---|---|---|
| protein_list | – | protein IDs (list of string) | Proteins to include in the graph. |
| ncbi_taxon_id | 9606 (human) | integer | NCBI taxon identifier. |
| sources | all sources | 'biogrid', 'string' | List of sources (databases) to retrieve the data from. |
| paginate | True | True, False | Whether to paginate the API calls for the sources that require it. |

## D.2 BioGRID

The following table shows the Graphein parameters for BioGRID. See also the BioGRID API.

| Parameter | Default | Valid values | Description |
|---|---|---|---|
| searchNames | True | True, False | If True, the interactor OFFICIAL_SYMBOL will be examined for a match with the protein list. |
| searchIds | True | True, False | If True, the interactor ENTREZ_GENE, ORDERED LOCUS and SYSTEMATIC_NAME (orf) will be examined for a match with the protein list. |
| searchSynonyms | True | True, False | If True, the interactor SYNONYMS will be examined for a match with the protein list. |
| searchBiogridIds | True | True, False | If True, the entries in the protein list will be compared to BIOGRID internal IDS which are provided in all Tab2 formatted files. |
| additionalIdentifierTypes | empty | string | Identifier types on this list are examined for a match with the protein list. |
| max | 10000 | integer | Number of results to fetch. Used for pagination. |

| | | | |
|---|---|---|---|
| interSpeciesExcluded | True | True, False | If True, interactions with interactors from different species will be excluded. |
| selfInteractionsExcluded | False | True, False | If True, interactions with one interactor will be excluded. |
| evidenceList | empty | Pipe-separated list of evidence codes from here | Any interaction evidence with its Experimental System in the list will be excluded from the results unless includeEvidence is set to true. |
| includeEvidence | False | True, False | If set to true, any interaction evidence with its Experimental System in the evidenceList will be included in the result |
| excludeGenes | False | True, False | If true, interactions containing genes in the input list will be excluded from the results. |
| includeInteractors | True | True, False | If true, in addition to interactions between genes on the input list, interactions will also be fetched which have only one interactor on the input list. |
| includeInteractorInteractions | False | True, False | If true, interactions between the input list's first order interactors will be included. |
| pubmedList | empty | string | Interactions will be fetched whose Pubmed Id is/ is not in this list, depending on the value of excludePubmeds. |
| excludePubmeds | False | True, False | If False, interactions with Pubmed ID in pubmedList will be included in the results; if 'true' they will be excluded. |
| htpThreshold | 20 | integer | Interactions whose Pubmed ID has more than this number of interactions will be excluded from the results. Ignored if excludePubmeds is False. |

| Parameter | Default | Valid values | Description |
|---|---|---|---|
| throughputTag | 'any' | 'low', 'high', 'any' | If set to 'low or 'high', only interactions with 'Low throughput' or 'High throughput' in the 'throughput' field will be returned. |

## D.3  STRING

The following table shows the Graphein parameters for STRING. See also the STRING API.

| Parameter | Default | Valid values | Description |
|---|---|---|---|
| network_type | 'functional' | 'functional', 'physical' | Network type: functional (default), physical. |
| add_nodes | 0 | integer | Adds a number of proteins to the network based on their confidence score, e.g., extends the interaction neighborhood of selected proteins to desired value. |
| show_query_node_labels | False | True or False | When available use submitted names in the preferredName column. |

# E  Parameters for gene regulatory networks

## E.1  General parameters

We download gene regulatory networks from TRRUST and RegNetwork. We build a directed graph where nodes are genes and attributed edges represent regulatory effects (activation, repression, or unknown).

| Parameter | Default | Valid values | Description |
|---|---|---|---|
| gene_list | – | gene symbols (list of string) | Genes to include in the graph. |

# F  Graph Construction Performance

We provide performance assessment performed on an AMD EPYC 7742 64-Core Processor. We use 20 PDBs (number os residues in brackets) with a mean length of 1,089.1 residues: 1n9u (10), 1j5l (30), 1ip0 (50), 1eod (100), 1agy (200), 1wzu (300), 1inp (400), 1a8h (500), 2ywe (600), 6cgm (700), 7lst (800), 4a7k (900), 4nab (1,000), 6mfw (1,210), 7edd (1519), 4f93 (1724), 6lqa (2059), 6r9t (2547), 4rh7 (3005), 5w1r (4128).

We observe construction time is marginal compared to download time for coarsened structural representations (Figure 1).

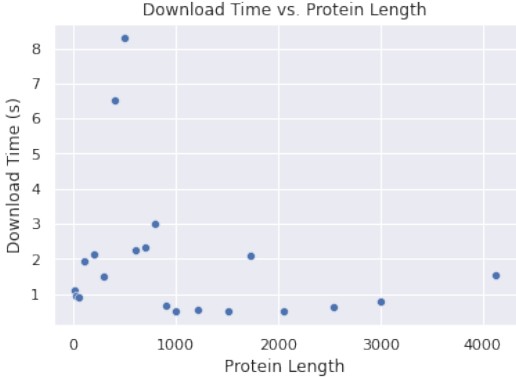

Figure 1: Download times for protein structures from the PDB.

### F.1 Cα Graphs

#### F.1.1 Peptide Bonds

We compute peptide bond graphs for the above list of 20 structures 50 times (i.e. 1,000 evaluations) in parallel (16 workers) and average across 7 runs to obtain an average construction time of 0.0071s per protein.

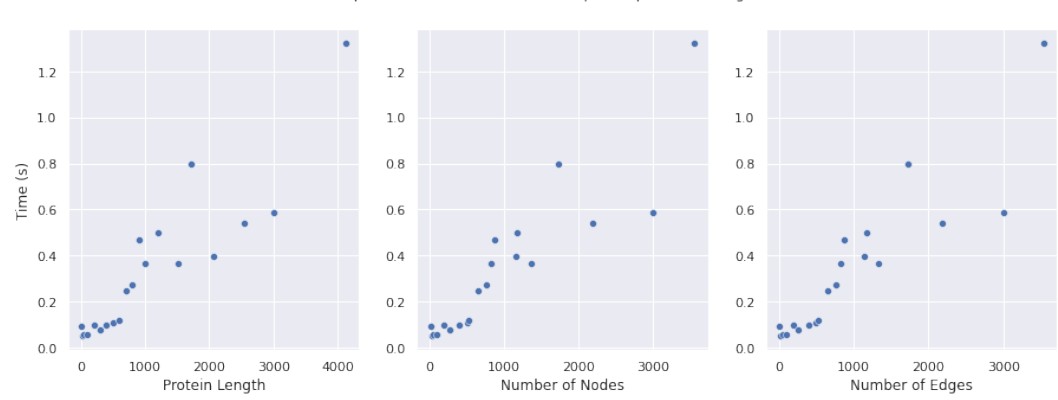

Figure 2: Graph construction times for peptide bond graphs.

#### F.1.2 Distance-based

We compute epsilon graphs ($\epsilon = 5$) for the above list of structures following the same procedure as F.1.1 to obtain an average construction time of 0.091s per protein.

#### F.1.3 Intramolecular

We compute protein structure graphs with Van der Waals interactions, Hydrogen bonds, salt bridges and disulfide interactions. We use the same procedure as in F.1.1 to obtain an average construction time of 0.117s per protein.

### F.2 Atomic Graphs

We compute atomic covalent bond graphs for the above list of 20 structures 5 times (i.e. 100 evaluations) in parallel (16 workers) and average across 7 runs to obtain an average construction time of 2.5s per protein.

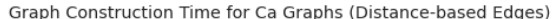

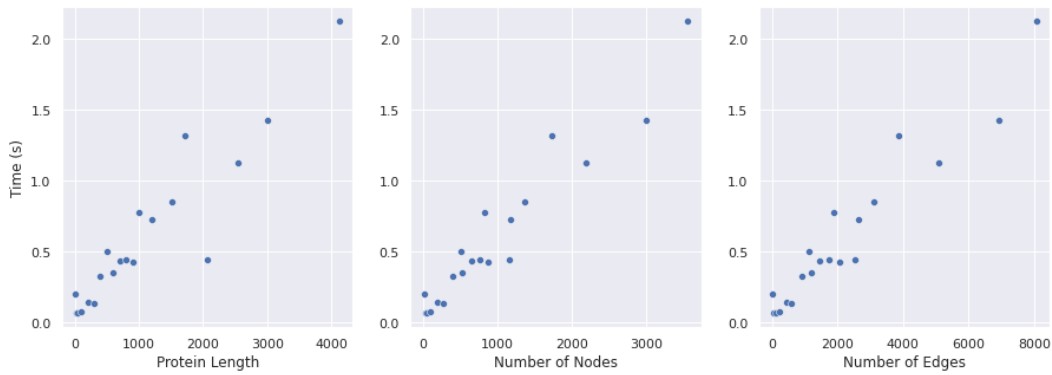

Figure 3: Graph construction times for distance-based edges.

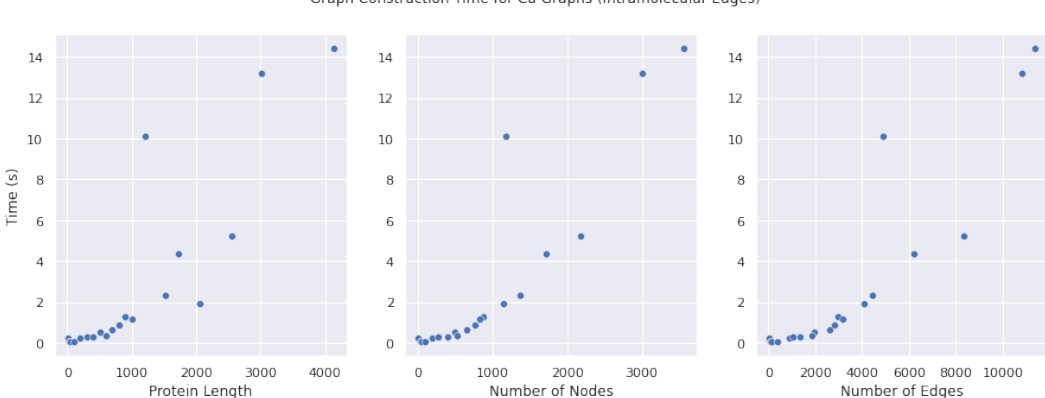

Figure 4: Graph construction times for intramolecular interaction graphs.

## G   Societal Impact

The potential risks of our library are minimal. A foreseeable hazardous application of our work is its use by nefarious actors to engineer harmful biomolecules, such as engineering toxins with greater efficacy. However, we believe these risks are minimal and shared across any developments in making computational design of biomolecules more accessible. We believe that utility of Graphein in the context of therapeutic development significantly outweighs these unlikely scenarios and anticipate our contribution to be a net force for social good and public health.

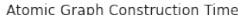

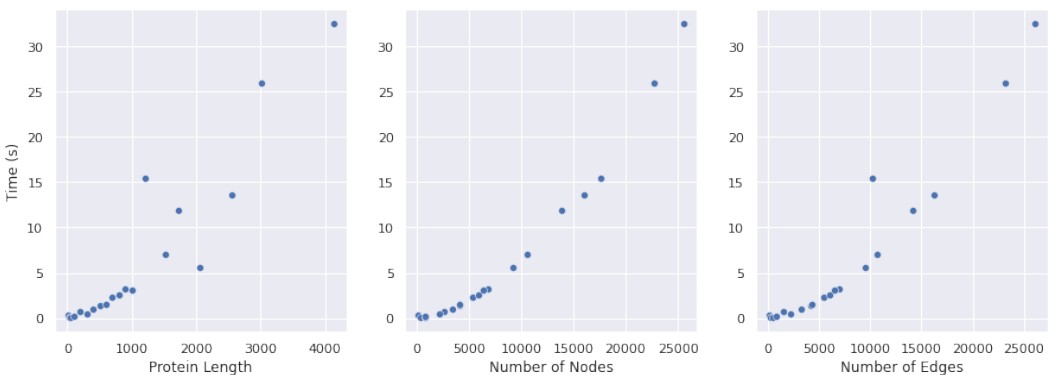

Figure 5: Graph construction times for atomic structure graphs.