# OpenReview forum: "Graphein - a Python Library for Geometric Deep Learning and Network Analysis on Biomolecular Structures and Interaction Networks"
_NeurIPS.cc/2022/Conference — NeurIPS 2022 Accept_

### Official Review · Reviewer_gZ1K · 2022-07-11

**Rating:** 6
**Confidence:** 2
**Soundness:** 4 excellent
**Presentation:** 3 good
**Contribution:** 4 excellent

**Summary:**

Graphein is a library for converting graph information contained in widely used bioinformatics databases to a machine learning friendly form. This allows machine learning researchers consistent and flexible preprocessing steps for structural and interaction graph data. As compared to previous widely used ML benchmarks and libraries which extract and collect fixed benchmark datasets, Graphein allows more customization to add new features as needed. This provides more low-level control for researchers to build and manipulate their own datasets beyond the standard datasets already supplied in tools like pytorch geometric, torch drug, and there are undoubtably researchers who are finding and will find Graphein useful for preparing biological structure and network data.

**Questions:**

Isn't this paper better suited for the datasets and benchmarks track? This paper is difficult to evaluate on the typical dimensions of NeurIPS main track: originality, quality, clarity, and significance.

Consistent and repeatable datasets with train and test splits have proven extremely useful in the broader ML field. Using these benchmarks, methods can be directly and easily compared with minimal task-specific knowledge. These benchmarks are relatively easy to standardize and agree upon in image and natural language tasks. However, in biology there are vastly greater possible tasks and more heterogenous data sources. Graphein provides a very flexible pre-processing pipeline. I wonder if the authors have any thoughts as to how to ensure consistency between as data sources and data pipelines change?


Minor comments:
Line 332: pre-rocessing  --> pre-processing

----------------------------
Post Response update:

Thank you to the authors for your detailed response. My concerns have been addressed and I have updated my score accordingly 4 --> 6. I believe additional discussion comparing Graphein to other molecule processing libraries would be helpful to answer the question "Why Graphein?" for ML practitioners, but overall a good contribution to ML infrastructure for biological graph dataset processing.

**Limitations:**

Yes

**Strengths And Weaknesses:**

Strengths:
* Flexible preprocessing pipeline that abstracts away some of the implementation details of bioinformatics datasets.
* Excellent and clearly written documentation and notebooks giving a high-level introduction to the library.
* Good visualization tools, particularly for protein structure graphs at various levels of granularity.
* Easy to install and provides docker containers for portability.

Weaknesses:
* The paper is mostly a summary of what the library is. I’m not sure I would ever recommend reading the paper over the documentation (of course this means the documentation is already excellent).
* There is limited comparison to the features provided in other related libraries. There are a number of tools that load and process geometric datasets, this paper could provide more information as to where Graphein fits in this larger ecosystem, what features it provides that other libraries do not, and why someone should use Graphein rather over custom preprocessing steps.
* There is no use / validation of datasets provided in the paper so it is difficult to evaluate how useful these particular datasets are in benchmarking methods.
* As this paper presents a python library for curating, visualizing, and preprocessing datasets, there is nothing novel in the traditional ML sense of new ideas or architectures to evaluate in this work.

---

> ### Author Response · Authors · 2022-07-28
> **Response to Reviewer gZ1K [1/2]**
>
> We thank the reviewer for their considered evaluation of our work. We are particularly glad the reviewer is of the opinion that graphein is and will continue to be useful to the research community.
>
> **The paper is mostly a summary of what the library is. I’m not sure I would ever recommend reading the paper over the documentation (of course this means the documentation is already excellent).**
>
> We thank the reviewer for their kind words about the quality of the documentation. We want the paper to serve as an introduction for life scientists to familiar problems that can be framed in the context of geometric deep learning or for experts in geometric deep learning who want to identify applications in life sciences.  We also seek to highlight the range of questions that researchers can address with our tool and the scope for integrating various modalities of biological data. We expect this is likely trivial to the reviewer given their expertise but hope they can recognise the utility of this framing. If the reviewer has any specific recommendations as to how we could better achieve this or any other ways in which it could be improved, we would, of course, be delighted to incorporate them into the revised manuscript.
>
> **There is limited comparison to the features provided in other related libraries. There are a number of tools that load and process geometric datasets, this paper could provide more information as to where Graphein fits in this larger ecosystem, what features it provides that other libraries do not, and why someone should use Graphein rather over custom preprocessing steps.**
>
> We thank the reviewer for raising this point and we would be delighted to expand on the differences in the revised manuscript. The major advantages that we see are the extensive documentation, ML framework agnosticisim, utilities for visualisation, processing, featurisation and dataset construction (including homology-based splitting), the ability to integrate disparate data sources, and the flexible style of implementation that enables users to reuse parts of and build on top of the library to address specific requirements. Furthermore, we believe that adopting graphein facilitates reproducibility and helps researchers build on previous work more easily.
>
> We would also like to raise the point that many of the contributors to Graphein are biologists (in particular structural biologists) and they have brought domain knowledge to the processing pipeline which other libraries lack by comparison. Structural data is complicated to work with in practice; for instance, structure files can provide alternate locations, multiple models, non-interesting ligands (such as crystallographic adjuvants), modified or non-standard amino acids or water molecules which need to considered in principled ways depending on the ultimate application. We have done this and therefore we greatly reduce the chances of unfamiliar users making pre-processing mistakes.
>
> Furthermore, we are aware of many users leveraging components of Graphein in their own custom processing steps [1-9]. This is enabled by the functional style of Graphein, its low-level API and permissive MIT license. The usage of the PyData stack also enables other types of applications and interoperability with many existing tools. For instance, we are aware of a group of users building interactive ML-driven applications on top of Graphein [10].
>
> **There is no use / validation of datasets provided in the paper so it is difficult to evaluate how useful these particular datasets are in benchmarking methods.**
>
> We appreciate the reviewers comments on this matter. We hope they serve as useful demonstrations of the ease with which novel datasets can be created and ingested with Graphein. Furthermore, most datasets that we are aware of focus on assessing global representations of structure (i.e. a prediction is made over the whole protein). We provide datasets that enable tasks that operate at a local level (i.e. predictions are made over regions of the protein structure). These datasets are typically harder to source as there is a much greater burden of annotation in their development. We agree benchmarking is important and we are developing a template repository which can be used for benchmarking purposes as well as a starting point for deep learning projects with engineering boilerplate taken care of.
>
> **As this paper presents a python library for curating, visualizing, and preprocessing datasets, there is nothing novel in the traditional ML sense of new ideas or architectures to evaluate in this work.**
> We appreciate this work is of a different focus and that focussing on benchmarking would blur the focus of the manuscript. We hope the reviewer can recognise utility in other areas of our work.

---

> > ### Author Response · Authors · 2022-07-28
> > **Response to Reviewer gZ1K [2/2]**
> >
> >
> > **Isn't this paper better suited for the datasets and benchmarks track? This paper is difficult to evaluate on the typical dimensions of NeurIPS main track: originality, quality, clarity, and significance.**
> >
> > We thank the reviewer for raising this concern. We carefully considered which venue was more suitable for our work. Ultimately we decided on the main track as the Call for Papers this year specifically calls for *“Infrastructure (e.g., datasets, competitions, implementations, libraries)”* and *“Machine Learning for Sciences (e.g. biology, physics, health sciences, social sciences)”* [11]. As the focus of our work is not on benchmarking we deemed the main track a better alignment.
> >
> > **Consistent and repeatable datasets with train and test splits have proven extremely useful in the broader ML field. Using these benchmarks, methods can be directly and easily compared with minimal task-specific knowledge. These benchmarks are relatively easy to standardize and agree upon in image and natural language tasks. However, in biology there are vastly greater possible tasks and more heterogenous data sources. Graphein provides a very flexible pre-processing pipeline. I wonder if the authors have any thoughts as to how to ensure consistency between as data sources and data pipelines change?**
> >
> > The reviewer raises a very interesting and important point about data and pipeline versioning. Ensuring reproducibility is a core aim of the library. Regarding version drift we have tried to address this in specific cases we are aware of. For instance, Graphein provides utilities for retrieving updated PDB identifiers as structures are made obsolete by new, higher-quality structures. The data sources Graphein makes use of are typically very mature databases with significant metadata about revisions and deposition dates that users can make use of. We also provide release notes with significant documentation about changes and previous releases are persistently available from PyPI. Ultimately, if strict versioning is required by a user we would expect them to store a copy of the raw data. We would also refer to the Dataset classes in Graphein which perform caching of raw and processed datasets which can be versioned appropriately and shared.
> >
> > **Minor comments: Line 332: pre-rocessing --> pre-processing**
> >
> > We thank the reviewer for spotting this. We will amend this in the revised manuscript.
> >
> > **References**
> >
> > [1] https://github.com/bougui505/misc/blob/1c832e55102235bab9b2a2583ca4ebaa716d6af2/python/graphein/protein.py
> >
> > [2] https://github.com/jipq6175/Interface_GN/blob/e7b949709101fa19ab8957e63d03455ec4580150/utils/data_utils.py
> >
> > [3] https://github.com/diamondspark/PSG-BAR/blob/28dd5cb42b4424aa55a9091eeca2a6984b862bd8/Preprocessing/GraphProcessing.py
> >
> > [4] https://github.com/cimranm/rAF2/blob/1713f87c1593637eb8eb0a2c44facb021d712a5b/load.py
> >
> > [5] https://github.com/biocad/CSM/blob/301bf5e2eea2601ce878302f7c3327081370de7c/create_CSM.py
> >
> > [6] https://github.com/jacksongrady/protein_folding_gsae/blob/0fb5e53cd1757ae22220a4ef83593bce66334d07/de_shaw_Dataset.py
> >
> > [7] https://github.com/adamcatto/MolFinDS/blob/94b8567eb18f415d51c93c5eb15a8ce8f05b1a40/data/preprocessing_helpers.py
> >
> > [8] https://github.com/BioinfoMachineLearning/ATOMRefine/blob/dd39125e61599debc62f238ee819e515caa8d021/covalent.py
> >
> > [9] https://github.com/BioinfoMachineLearning/DeepRefine/blob/5e2ea4de4aa3e8510c9e74d83f7516d111e31fc9/project/utils/deeprefine_utils.py
> >
> > [10] https://github.com/cimranm/pomegranate
> >
> > [11] https://nips.cc/Conferences/2022/CallForPapers

---

### Official Review · Reviewer_wsXa · 2022-07-12

**Rating:** 7
**Confidence:** 3
**Soundness:** 3 good
**Presentation:** 3 good
**Contribution:** 3 good

**Summary:**

This work introduces a Python library, Graphein, for transforming raw data including biomolecular structures and interactions from widely used bioinformatics databases into the datasets for geometric deep learning. Graphein supports a broad selection of data sources including small molecules, proteins, RNA, and biological interaction networks. Besides, Graphein provides several machine learning utilities for users to better work with the data.

**Questions:**

Currently, only a limited number of datasets are constructed in Graphein. Will more datasets that better cover each of the supported data types/sources be designed in the future?

**Limitations:**

To attract more researchers to use Graphein in their work, the results of baseline models can be collected and provided together with the datasets made by Graphein for comparison.


**Strengths And Weaknesses:**

- Originality: Although there are existing tools for building biomolecular graphs, they all exhibit limited utilities for geometric deep learning. The proposed tool covers a broader selection of bioinformatics data and provides more utilities for the transformed graph-structured data.
- Quality: The proposed library provides convenient structure retrieval and graph creation tools. More empirical results based on the library need to be collected and shown.
- Clarity: The paper is well written and the proposed tool is clearly introduced.
- Significance: Geometric deep learning in the field of bioinformatics is getting more and more popular these years. Although there are several tools for transforming bioinformatics data to graph-structured data, they only support a small selection of data sources, and the transformed data can only be directly used by limited geometric deep learning libraries. Given that the proposed tool provides wider support for both data sources and downstream libraries, many researchers would be interested in using this tool for preparing graph-structured data for various tasks.

---

> ### Author Response · Authors · 2022-07-28
> **Response to Reviewer wsXa**
>
> We thank the reviewer for their considered evaluation of our work. We are very glad that the reviewer is of the opinion that our work would be of interest and value to the research community.
>
> **Currently, only a limited number of datasets are constructed in Graphein. Will more datasets that better cover each of the supported data types/sources be designed in the future?**
>
> Yes, we acknowledge the limited number of datasets available at present. We are actively working on further curation of suitable datasets that encompass the range of tasks researchers may like to pursue with our tool. We anticipate that further engagement, dialogue and contributions from the community will accelerate this. We would like to highlight the ease with which new datasets can be ingested with Graphein. For example, we provide two example notebooks showing how datasets in the Therapeutic Data Commons [1] can be rapidly ingested. We are actively collaborating with the developers of the Therapeutic Data Commons to provide further integration of our tools.
>
> Example 1: Predicting small molecule toxicity
> https://anonymous.4open.science/r/graphein-3472/notebooks/molecule_model_tutorial_tox.ipynb
>
> Example 2: Predicting antibody developability
> https://anonymous.4open.science/r/graphein-3472/notebooks/tdc_developabiXXXX-39ty.ipynb
>
> **To attract more researchers to use Graphein in their work, the results of baseline models can be collected and provided together with the datasets made by Graphein for comparison.**
>
> We agree with the reviewer that collecting baseline results will be an important contribution. It is very much on our roadmap with high priority. We have reflected on the best way to do this and we would greatly appreciate the opinion of the reviewer. We are currently developing a highly configurable ML project template repository with many standard baselines and datasets implemented, with engineering boilerplate such as metrics and logging taken care of. We will make this available to the community before the camera-ready deadline.
>
> **References**
>
> [1] [Therapeutics Data Commons: Machine Learning Datasets and Tasks for Drug  Discovery and Development](https://arxiv.org/pdf/2102.09548)

---

### Official Review · Reviewer_qxsn · 2022-07-12

**Rating:** 8
**Confidence:** 5
**Soundness:** 4 excellent
**Presentation:** 4 excellent
**Contribution:** 4 excellent

**Summary:**

This paper presents "Graphein", a tool for the pre-processing and overall data wrangling of biological networks for machine learning tasks.

**Questions:**

- The documentation seems to be generated from python notebooks, and the output includes a lot of debug information. Could this be made "toggleable"? It will improve with initial reads of the reference, as (at a first glance) the output looks like an error.

**Limitations:**

Considering the quality of Graphein and the existing quality of other well-established pieces of code oriented for computational biology, I consider the limitations in the documentation a minimal issue for Graphein.

**Strengths And Weaknesses:**

The tool is very well documented and the code seem to be professional-grade.

While the documentation in it's current version is very useful, it feels incomplete. This is mitigated by the authors providing a set of easy to follow and high quality set of examples, provided as python notebooks.

The single "weakness" of this package in my opinion is the lack of a manual for the "advanced API" (although this reasonable, considering the audience of such and API are expected to understand the inner workings of Graphein to use it adequately).

I've managed to run ALL the examples successfully in a couple of days, and had minimial issues that could be resolved with a quick search on the documentation or reading the next section of the manual.

---

> ### Author Response · Authors · 2022-07-28
> **Response to Reviewer qxsn**
>
> We thank the reviewer for their considered response and for taking the time to engage with all of the examples and features of the library. We are also very pleased to hear the reviewers comments on the quality of the code and documentation.
>
> **The single "weakness" of this package in my opinion is the lack of a manual for the "advanced API" (although this is reasonable, considering the audience of such an API are expected to understand the inner workings of Graphein to use it adequately)**
>
> We are grateful to the reviewer for drawing our attention to this. We will include richer documentation and tutorial for the low-level API in our next release. Usability is a primary focus of ours.
>
> **The documentation seems to be generated from python notebooks, and the output includes a lot of debug information. Could this be made "toggleable"? It will improve with initial reads of the reference, as (at a first glance) the output looks like an error.**
>
> We thank the reviewer for raising this point. Yes, we will refactor the logging to make the documentation cleaner in our next release.
>
> **I've managed to run ALL the examples successfully in a couple of days, and had minimal issues that could be resolved with a quick search on the documentation or reading the next section of the manual.**
>
> We thank the reviewer for making the effort to run the examples. If the reviewer recalls any of the minimal issues they encountered we would be delighted to add further clarity where needed.

---

### Official Review · Reviewer_VXGc · 2022-07-18

**Rating:** 6
**Confidence:** 3
**Soundness:** 2 fair
**Presentation:** 3 good
**Contribution:** 2 fair

**Summary:**

This paper provides a python library on modeling protein-protein interaction for the field of target discovery and drug design.
It includes commonly used database on both structural data and interaction data. Also, for neural network and modeling, it provides interfaces on different scale level for atom level to protein/DNA/RNA level. The code is open-access, which could be helpful for the drug community.

**Questions:**

- It is well-known that some databases could be extremely large. Could you provides the performance running on large-scale databases.

- do you plan to provide AI baselines and benchmark different methods on the open databases?

**Limitations:**

- my major concern is the scale and efficiency for this toolbox, especially for large-scale data.

**Strengths And Weaknesses:**

- Strengths: this paper gives a brief overview of widely used databases and modeling interface for protein interface and mesh. The python code is open access. This tools could be beneficial for communities from both target discovery and drug design.

- Weaknesses: it could be better if this toolbox could provide common methods and make benchmarks on these standard databases.

---

> ### Author Response · Authors · 2022-07-28
> **Response to Reviewer VXGc**
>
> We thank the reviewer for their constructive comments on our work. We are glad they recognise the benefit of our tool for the computational drug discovery and life science communities.
>
> **It is well-known that some databases could be extremely large. Could you provide the performance running on large-scale databases?**
>
> The reviewer raises a very important point regarding the scale of data users may wish to work with. We have performed benchmarking of various types of graph construction and conclude that this is not a major limitation. We will be pleased to include these results in the revised manuscript. Our analysis is available at the following URL: ​​https://anonymous.4open.science/r/graphein-runtime-6E62/time.ipynb
>
> Additionally, Graphein provides utilities for parallelizing the processing of large datasets and the provided Dataset classes perform caching of processed data. Furthermore, we have reports from users successfully using Graphein to work with datasets of over 300,000 structures and have successfully used Graphein ourselves to process datasets in the order of 1,000,000 structures (all proteins in AF2 Database at time of writing and the PDB). Typically, the rate limiting step we have observed is actually the API from the underlying database sources which we cannot do much to alter and will be faced by any workflow retrieving data therefrom. We are aware of opportunities to optimize the library further and are actively doing so as part of our development roadmap. We hope this addresses the reviewers' concerns sufficiently.
>
> **Do you plan to provide AI baselines and benchmark different methods on the open databases?**
>
> Yes, we think this will be a very important contribution and natural extension of our work. It is very much on our roadmap with high priority. We have reflected on the best way to do this to provide the most impact to the community and we would greatly appreciate the opinion of the reviewer. We are currently developing a highly configurable ML project template repository with many standard baselines and datasets implemented and engineering boilerplate such as metrics and logging taken care of. We will make this available to the community before the camera-ready deadline.

---

### Meta-Review · Area_Chair_4kq8 · 2022-08-26

**Recommendation:** Accept
**Confidence:** Certain

**Metareview:**

The paper describes a python library called Graphein, for working with biomolecules. The manuscript provides a bridge between life scientists and machine learners, describing the high level concepts that will facilitate a meaningful interaction between the two fields by suitable use of the Graphein library. The library provides programmatic interfaces to query bioinformatics databases, and also enables the use of popular geometric deep learning libraries.

Four reviewers carefully considered the paper, and the associated library, and they unanimously agree that the work provides an excellent contribution to the machine learning literature. It gives me great pleasure that the new directions of software in the NeurIPS community has attracted high quality submissions such as this. Therefore I recommend this paper for acceptance at NeurIPS 2022. Congratulations!

**Award:**

No

---

### Decision · Program_Chairs · 2022-09-14

Accept